# Exploring View Sampling Strategy in Novel View Synthesis from Causal Perspectives

## Abstract

Neural Radiance Fields (NeRF) has shown promising performance in synthesizing high-quality and realistic images. However, it often relies on a large amount of high-quality training data. Instead of extensively sampling training samples to cover various details of scenes, a series of works have studied how to utilize prior knowledge to achieve high-quality novel view synthesis with limited training samples. However, these methods have not explored the essence of this problem, which is how to get the optimal training set under limited view inputs. ActiveNeRF proposes a method based on an active learning scheme that evaluates the reduction of uncertainty given new inputs, selects samples that provide the maximum information gain, and adds them to the existing training set. However, it evaluates information gain by the variance change of prior and posterior distributions, which is merely an intuitive method and may result in unreliable results. We revisit the view sampling strategies from a causal perspective and achieve efficient sampling without requiring the ground truth of invisible samples. We also propose a new theoretical framework for the sampling problem in NeRF. We analyze how to obtain the optimal sampling strategy based on our framework. Experiments show that our conclusion can not only guide sampling but also help us design the regularization term for general NeRF.

## 1 Introduction

NeRF Mildenhall et al. (2021) (Neural Radiance Fields) is a state-of-the-art technique in computer graphics and computer vision that enables the generation of highly detailed and photorealistic 3D reconstructions of scenes or objects from 2D images. NeRF represents a scene or object as a 3D volume, where each point in the volume corresponds to a 3D location and is associated with a color and opacity. The key idea behind NeRF is to learn a deep neural network that can implicitly represent this volumetric function, allowing the synthesis of novel views of the scene from arbitrary viewpoints.

Although NeRF can synthesize high-quality and realistic images, it often relies on a large amount of high-quality training data Yu et al. (2021b). The performance of NeRF drastically decreases when the number of training data is reduced. Previous NeRF generally achieved high-quality synthesis by extensively sampling training samples to cover various details of scenes. Additionally, a series of works Niemeyer et al. (2022); Yang et al. (2023); Yu et al. (2021b); Jain et al. (2021); Zhou & Tulsiani (2022); Deng et al. (2022), have studied how to utilize prior knowledge to achieve high-quality novel view synthesis with limited training samples, such as leveraging additional sources of prior information, designing more efficient training algorithms, and incorporating domain knowledge to enhance generalization. However, these methods have not explored the essence of this problem, which is how to get the optimal training set under limited view inputs. Exploring this problem can not only enable efficient sampling, avoiding data redundancy but also guide us in designing prior regularization terms for novel view synthesis with limited training samples.

The most relevant work related to our research is ActiveNeRF Pan et al. (2022), which aims to model 3D scenes using limited input resources. First, it incorporates uncertainty estimation into the NeRF model to ensure robustness under a few observations. Then, it proposes a method based on an active learning scheme that evaluates the reduction of uncertainty given new inputs, selects samples that provide the maximum information gain and adds them to the existing training set. This approach allows for improving the quality of novel view synthesis with minimal additional resource investment.

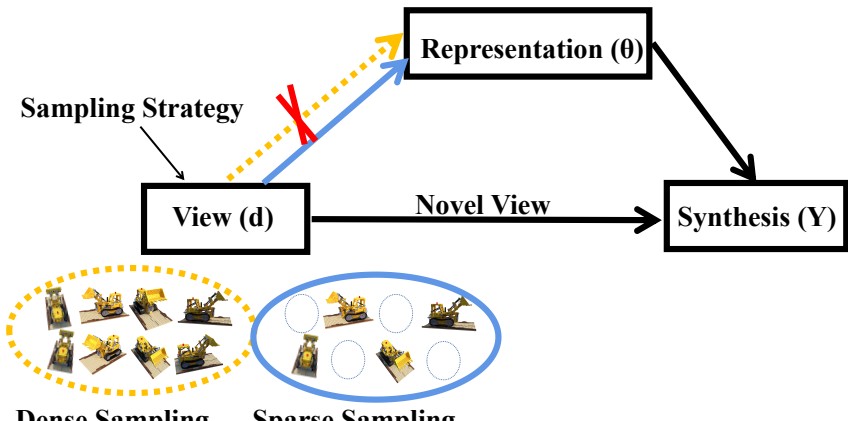

Figure 1: Novel view synthesis via NeRF from the casual perspective. $d$ represents the candidate viewpoints, $\theta$ represents the learned parameters (representations) of the NeRF from input samples, and $Y$ represents the synthesized novel images rendered by the NeRF from novel viewpoints. During sparse view sampling, due to the scarcity and randomness of the samples, the learned $\theta$ is often unstable, indicating the presence of an edge $d \rightarrow \theta$. However, compared to sparse view sampling, dense view sampling implies that the NeRF is more likely to fit the entire real scene, resulting in a stable and optimal $\theta$ where the edge $d \rightarrow \theta$ does not exist. Our goal is to learn stable representations $\theta$ from sparse samples by removing $d \rightarrow \theta$.

However, it evaluates information gain by the variance change of prior and posterior distributions, which is merely an intuitive method and may result in unreliable results. Moreover, it takes a long time to select a new view, which significantly increases training time.

Our key observation is that there is a potential relationship between causal representation learning and novel view synthesis. Causal representation learning focuses on understanding the causal relationships and underlying factors that drive the data generation process. For example, we can explore the causality between taking medication and recovering from a cold. In NeRF, we can regard the visibility of viewpoints as a treatment variable, which essentially represents a view-sampling strategy for sparse inputs. We can also investigate the causality of different view sampling strategies on the novel view synthesis. Our goal is to achieve stable and excellent novel view synthesis results under different view sampling strategies.

In this paper, we first revisit the novel view synthesis via a casual perspective. Inspired by causal representation learning, we reformulate the task loss by ITE(Identification Treatment Effect). Then we propose a framework that theoretically supports the main idea. We derived the upper bound of the ITE loss, which can be divided into three terms, each modeling the problem from a different perspective. The fitting term requires the optimal network parameters under a certain view sampling strategy to support correct fitting to the ground truth. The consistency term requires that, under the current sampling strategy, the optimal network parameters obtained from visible views and invisible views are as consistent as possible. This actually means that the distributions of visible views and invisible views should be consistent. The uniformity term requires the sampled views to be distributed as evenly as possible. This upper bound can guide us on how to capture new views to supplement the existing training set, which brings the most information gain. Moreover, the conclusion can not only guide sampling but also help the design of regularization terms because these properties are necessary conditions for an outstanding NeRF.

Finally, we have validated our conclusion experimentally. First, we validate whether this conclusion can guide sampling, followed by ActiveNeRF Pan et al. (2022). We provide several initial images and supplement new viewpoints to evaluate whether the sampling strategy can achieve the best performance with the minimum number of samples. The experiments demonstrate that based on our sampling strategy, the best performance can be achieved by introducing the same number of new viewpoints. Additionally, we design a regularization term based on this conclusion to constrain the training of general NeRF, and the experiments show that these regularization terms can significantly improve performance. Both experiments demonstrate that our framework is general and effective.

## 2 RELATED WORK

**Causal representation learning** To address the issue of uneven distribution between the treatment and control groups (e.g., out-of-distribution), causal representation learning (CRL) Shalit et al. (2017); Johansson et al. (2016) is an essential framework in addition to traditional weighted sampling methods (e.g., IPW estimator Austin & Stuart (2015)). In this framework, we learn a common representation for the covariates of both the treatment and control groups, which aims to have a similar distribution between the two groups.

**Active learning.** Active learning is a special case of machine learning in which a learning algorithm can actively seek user (or another information source) input to label new data points with desired outputs. It has been extensively explored across diverse computer vision tasks Yi et al. (2016); Sener & Savarese (2017); Fu et al. (2018); Zolfaghari Bengar et al. (2019). ActiveNeRF Pan et al. (2022) is the first approach to incorporate an active learning scheme into the NeRF optimization pipeline. We adopt this active learning pipeline for novel view synthesis. In contrast to ActiveNeRF's focus on modeling information gain through uncertainty reduction, our approach aims to explore upper bounds for sampling from a causal perspective.

**Few-shot Novel View Synthesis.** NeRF Mildenhall et al. (2021) has become one of the most important methods for synthesizing new viewpoints in 3D scenes Xiangli et al. (2021); Fridovich-Keil et al. (2022); Takikawa et al. (2021); Yu et al. (2021a); Tancik et al. (2022); Hedman et al. (2021). A growing number of recent works have studied few-shot novel view synthesis via NeRF Wang et al. (2021); Meng et al. (2021); Niemeyer et al. (2022); Kim et al. (2022); Wang et al. (2023). First, diffusion-model-based methods use generative inference as supplementary information. NeRDi Deng et al. (2022) proposes a single-view NeRF synthesis framework with general image priors from 2D diffusion models. SparseFusion Zhou & Tulsiani (2022) tries to distill a 3D consistent scene representation from a view-conditioned latent diffusion model. Second, some methods additionally extrapolate the scene's geometry and appearance to a new viewpoint. DietNeRF Jain et al. (2021) introduces additional semantic consistency loss between observed and unseen views based on pre-trained CLIP Radford et al. (2021) models. Third, some methods utilize different regularization terms to avoid overfitting. RegNeRF Niemeyer et al. (2022) regularizes the geometry and appearance of patches rendered from unobserved viewpoints. FreeNeRF Yang et al. (2023) propose two regularization terms to regularize the frequency range of NeRF's inputs and to penalize the near-camera density fields. In our work, we aim to unify these specific previous works into our more general framework.

## 3 PRELIMIARIES

NeRF Mildenhall et al. (2021) is a neural rendering framework that provides a mapping function f from 3D space coordinates $x$ and viewing directions $d$ to volume density $\sigma$ and color $c$. Specifically, NeRF achieves this by establishing a mapping function from the input 3D space coordinates and viewing directions to the output color and transparency via a deep neural network referred to as the NeRF model. The model is trained from a series of images and can subsequently be utilized for synthesizing images from new viewpoints and conducting 3D scene reconstruction. Specifically, we have $f_{\boldsymbol{\theta}} : (\gamma(x), \gamma(\boldsymbol{d})) \mapsto (\sigma, \mathbf{c})$. Here $\gamma$ is a positional encoding. Via casting a ray $r(t) = o + td$, each pixel color can be denoted as $\hat{C}(\boldsymbol{r}) = \sum_{i=1}^{N} c_i^*$ Wang et al. (2023), where $c^* = T_i(1 - e^{-\sigma_i \delta_i})\boldsymbol{c}_i$, where $T_i = e^{-\sum_{j=1}^{i-1} \sigma_j \delta_j}$, $\delta_i = t_i - t_{i-1}$.

Here $t_i$ is selected from the partitions (we partition $[t_n, t_f]$ into $N$ points $[t_1, t_2, t_3, ...t_N]$). We aim to estimate the loss $\mathcal{L}_{nerf} = \sum_{r \in R} \|\hat{C}(\boldsymbol{r}) - C(\boldsymbol{r})\|^2$. Here $\hat{C}(\boldsymbol{r})$ is the image to be rendered via $n$ samples, while $C(\boldsymbol{r})$ is the ground-truth picture.

## 4 METHODOLOGY

### 4.1 MOTIVATION: THE RELATIONSHIP BETWEEN CRL AND NERF

Inspired by Figure. 1, the core difference between dense-sample learning and sparse-sample learning is whether the representation $\boldsymbol{\theta}$ is well-defined or not. Inspired by this, we aim to transform the

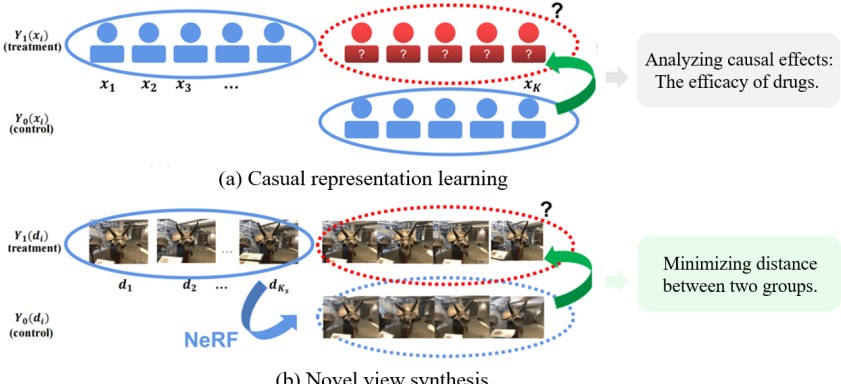

Figure 2: **(a)** CRL process: we rely on a random sampling strategy to divide the population into two groups that receive treatment (i.e., $Y(1)$) or act as a control group (i.e., $Y(0)$). We hope to overcome the difficulty of uneven distribution of the covariate $X$ in the two population groups by means of representation constraints and recover as true a $Y(1)$ as possible from the $Y(0)$ group. **(b)** Revisit the novel view synthesis through CRL: we consider the process of selecting $K_s$ sparse views among $K_d$ sampled views as treatment, and the remaining $(K_d - K_s)$ views as the control group. At the same time, we use the real-world sample/image synthesized by NeRF($\boldsymbol{\theta}$) as the information of treatment and control groups, respectively. Note that the difference between (b) and (a) is that we aim to minimize the distance between NeRF's output and its ground-truth, which is same as the potential outcomes from CRL.

problem of learning stable representations $\boldsymbol{\theta}$ from sparse samples into the problem of removing $d \to \boldsymbol{\theta}$. This ensures that the sampling pattern does not affect the final results.

The most similar idea could be traced back to the field of causal representation learning. Among which the widely practised work Shen et al. (2021); Wang et al. (2020); Zhang et al. (2020); Tang et al. (2020); Yang et al. (2021) can be broadly divided into two categories: a) using sample weighting methods to make the treated/untreated samples as independent as possible ($d \nrightarrow \boldsymbol{\theta}$) Shen et al. (2021), and b) manually fitting $\mathbb{P}(Y|do(d))$ instead of $\mathbb{P}(Y|d)$ using backdoor criterion Wang et al. (2020); Zhang et al. (2020); Tang et al. (2020); Yang et al. (2021). However, these methods often face unavoidable challenges, such as high computational complexity and inaccurate estimation. In this paper, we revisit and establish the connection between NeRF and CRL.

We compared the differences and similarities between causal representation learning and novel view synthesis. As shown in Figure 2, causal representation learning can effectively handle the imbalance distribution between the treatment group and the control group, for example, inferring the outcomes of another 5 individuals who did not take medication based on the outcomes of 5 individuals who did take medication. For novel view synthesis, we consider visible views and invisible views as the treatment group and control group respectively. We can predict the potential outcomes of the control group under treatment. Instead of analyzing causal effects, our goal is to make the results of control group samples as similar as possible to their results under treatment. This means that the visibility of the current view does not affect the results, achieving the decoupling of sampling patterns and results.

Since our goal is to make the results of samples from the control group as similar as possible to their results under treatment, we constructed a loss function with the variables of the sampling strategy and the optimal NeRF network parameters under it. We derived the upper bound of this loss, which can be divided into three terms, each modeling the problem from a different perspective. The fitting term requires the optimal network parameters under the current sampling strategy to support correct fitting to the ground truth. The consistency term requires that, under the current sampling strategy, the optimal network parameters obtained from visible views and invisible views are as consistent as possible. This actually means that the distributions of visible views and invisible views should be consistent. The uniformity term requires the sampled views to be distributed as evenly as possible. This upper bound can guide us how to capture new views to supplement the existing training set, which brings the most information gain. Moreover, the conclusion can not only guide sampling but also help the design of regularization terms because these properties are necessary conditions for an outstanding NeRF.

## 4.2 FRAMEWORK

NeRF is designed to reconstruct the color of a scene by utilizing $K_d$ dense views and enables the synthesis of new viewpoints. In order to investigate the sampling problem in NeRF, our main focus is on how to achieve optimal results with limited viewpoints. Our goal is to find an optimal sampling strategy for the training set.

We assume there are only $K_s \ll K_d$ samples to repeat the aforementioned steps, seeking to achieve optimal results. It is worth mentioning that the conclusion drawn from this setting can easily extend to others. So we start from considering a general setting following ActiveNeRF Pan et al. (2022):

*Given only $K_s$ corresponding photos as sparse sample inputs among $K_d$ total potential views, how should we achieve the best novel view synthesis?*

Noticing that within $K_s$ sparse views $\{d_1, d_2, ...d_{K_s}\}$ among the total views $K$, each image is denoted as $\{c(d_1), c(d_2), ...c(d_{K_s})\}$. We define the selection of a particular viewpoint as the treatment group (input, $t_i = 1$) and the non-selection of a viewpoint as the control group (no input, $t_i = 0$), and then $t = \{t_1, t_2, ...t_{K_d}\}, \|\{i : t_i = 1\}\| = K_s$. Being aligned with the framework of causal representation learning, we denote the potential outcome framework Rubin (2005) as follows. For each view $d_i$, we have two choices: 1) directly capturing the ground-truth scene or 2) reconstructing the uncaptured scenes:

- Capture the Ground-truth scenes: $Y_1(d_i) \overset{def}{=} c_i(d_i)$. We consider the act of obtaining sparse samples as the treatment condition and obtaining the ground truth image as the outcome.

- Reconstruct the uncaptured scenes: $Y_0(d_i) = \hat{c}_i(d_i) \overset{def}{=} f_{\theta_t}(d_i)$, where $\theta_t$ is trained via sparse samples $\{c(d_j), d_j\}_{t_j=1}$: $\theta_t = \arg\min_{\theta} \sum_{i:t_i=1} \mathcal{L}(f_\theta(d_i), c_i(d_i))$. $\theta_t$ is the optimal NeRF network parameters under sampling strategy $t$. Then the novel view (the estimand $\hat{c}(d_i)$ can be reconstructed.

For instance, if we receive the sparse view $K_s$ as $t = \{1, 1, 1, ....0, 0\}$, then the corresponding potential outcome Rubin (2005) is $\{Y_1(d_1), Y_1(d_2), Y_1(d_3), ...Y_0(d_{K_s-1}), Y_0(d_{K_s})\} = \{c_1(d_1), c_2(d_2), c_3(d_3), ...\hat{c}_{K_s-1}(d_{K_s-1}), \hat{c}_{K_s}(d_{K_s})\}$. Hence the reconstruction of the image can be denoted as follows in each iteration:

$$Y_i = t_i Y_1(d_i) + (1 - t_i)Y_0(x_i) = t_i c_i(d_i) + (1 - t_i)f_{\theta_t}(d_i). \tag{1}$$

Notice that the value of the control group is related to the treatment assignment $t$. Moreover, we assume that all sources of randomness in this problem stem from our specific sparse sampling strategy. Following the definition Johansson et al. (2016), we use $P^F$ to denote the factual distribution of sparse sampling $t = \{t_1, t_2, ...t_{K_s}\}$, and $\hat{P}^F$ to denote its empirical distribution. Analogously, we use $P^{CF}$ to denote the counterfactual distribution of sparse sampling $1 - t = \{1 - t_1, 1 - t_2, ...1 - t_{K_d}\}$, and $\hat{P}^{CF}$ denotes its empirical counterfactual distribution.

Recall that the loss of NeRF can be denoted as $\mathcal{L}_{NeRF}(\theta; t) = \sum_{[K_s]} [\mathcal{L}(f_\theta(d_i), c_i(d_i))\mathbb{I}(t_i = 1)]$. Under the sampling strategy of sparse views, the loss is set as

$$\mathcal{L}(\theta_t) = \mathbb{E}_{t \sim P^F} \mathcal{L}_{NeRF}(\theta, t) = \mathbb{E}_{t \sim P^F} \sum_{[K_s]} [\mathcal{L}(f_{\theta_t}(d_i), c_i(d_i))\mathbb{I}(t_i = 1)]. \tag{2}$$

Notice that the independent variable here is $\theta_t$, $\theta_t$ is a function of $t$. $\theta_t$ is the optimal NeRF network parameters under sampling strategy $t$. We aim to transfer the form of the loss function. Following the definition in Johansson et al. (2016), we denote the factual loss and counterfactual loss:

$$\mathcal{L}_{PF}(\theta_t) = \mathbb{E}_{t \sim P^F}[\mathcal{L}(f_{\theta_t}(d_i), Y_i)],$$
$$\mathcal{L}_{PCF}(\theta_t) = \mathbb{E}_{t \sim P^{CF}}[\mathcal{L}(f_{\theta_t}(d_i), Y_i)]. \tag{3}$$

Inspired by Shalit et al. (2017); Johansson et al. (2016), the sum of *factual loss* and *counterfactual loss* can bound $\mathcal{L}(\theta_t)$. Notice that (we defer the details to the Appendix) $\mathcal{L}(\theta_t) \leq K_1[\mathbb{E}_{t \sim P^F}\mathcal{L}(f_{\theta_t}(d_i), Y_i) + \mathbb{E}_{t \sim P^{CF}}\mathcal{L}(f_{\theta_t}(d_i), Y_i)] - K_2$, where $K_1, K_2$ are two bounded constants. We can transfer our attention to consider $\min[\mathbb{E}_{t \sim P^F}\mathcal{L}(f_{\theta_t}(d_i), Y_i) + \mathbb{E}_{t \sim P^{CF}}\mathcal{L}(f_{\theta_t}(d_i), Y_i)]$.

Namely we directly minimize the sum of factual loss and counterfactual loss. Notice that

$$
\begin{aligned}
&\min \mathbb{E}_{\boldsymbol{t}\sim PF}\mathcal{L}(f_{\boldsymbol{\theta_t}}(\boldsymbol{d}_i), Y_i) + \mathbb{E}_{\boldsymbol{t}\sim PCF}\mathcal{L}(f_{\boldsymbol{\theta_t}}(\boldsymbol{d}_i), Y_i)\\
&\leq \mathbb{E}_{\boldsymbol{t}\sim PF}\mathcal{L}(f_{\boldsymbol{\theta}_t^{CF}}(\boldsymbol{d}_i), Y_i) + \mathbb{E}_{\boldsymbol{t}\sim PCF}\mathcal{L}(f_{\boldsymbol{\theta}_t^{CF}}(\boldsymbol{d}_i), Y_i)\\
&\leq \mathbb{E}_{\boldsymbol{t}\sim PF}\mathcal{L}(f_{\boldsymbol{\theta}_t^{F}}(\boldsymbol{d}_i), Y_i) + \mathbb{E}_{\boldsymbol{t}\sim PCF}\mathcal{L}(f_{\boldsymbol{\theta}_t^{CF}(\boldsymbol{d}_i), Y_i) + \mathbb{E}_{\boldsymbol{t}\sim PF}\mathcal{L}(f_{\boldsymbol{\theta}_t^{CF}}(\boldsymbol{d}_i), f_{\theta_t^F}(\boldsymbol{d}_i))}
\end{aligned}
\tag{4}
$$

We define a knock-off $\bar{Y}$ to show its expectation: $\bar{Y}_i = t_i c_i(\boldsymbol{d}_i) + (1-t_i)\mathbb{E}_{\boldsymbol{t}\sim PF}f_{\boldsymbol{\theta_t}}(\boldsymbol{d}_i)$. Therefore

$$
\begin{aligned}
\mathbb{E}_{\boldsymbol{t}\sim PF}\mathcal{L}(f_{\boldsymbol{\theta}_t^F}(\boldsymbol{d}_i), Y_i) &\leq \mathbb{E}_{\boldsymbol{t}\sim PF}\mathcal{L}(f_{\boldsymbol{\theta}_t^F}(\boldsymbol{d}_i), \bar{Y}_i) + \mathbb{E}_{\boldsymbol{t}\sim PF}\mathcal{L}(Y_i, \bar{Y}_i).\\
\mathbb{E}_{\boldsymbol{t}\sim PCF}\mathcal{L}(f_{\boldsymbol{\theta}_t^{CF}}(\boldsymbol{d}_i), Y_i) &\leq \mathbb{E}_{\boldsymbol{t}\sim PCF}\mathcal{L}(f_{\boldsymbol{\theta}_t^{CF}}(\boldsymbol{d}_i), \bar{Y}_i) + \mathbb{E}_{\boldsymbol{t}\sim PCF}\mathcal{L}(Y_i, \bar{Y}_i).
\end{aligned}
\tag{5}
$$

On the one hand, we have:

$$
\begin{aligned}
&\mathbb{E}_{\boldsymbol{t}\sim PCF}\mathcal{L}(f_{\boldsymbol{\theta}_t^{CF}}(\boldsymbol{d}_i), \bar{Y}_i)\\
&= \mathbb{E}\mathbb{P}(t_i=0)\mathcal{L}(f_{\boldsymbol{\theta}_t^{CF}}(\boldsymbol{d}_i), c_i(\boldsymbol{d}_i)) + \mathbb{E}\mathbb{P}(t_i=1)\mathcal{L}\left(f_{\boldsymbol{\theta}_t^{CF}}(\boldsymbol{d}_i), \mathbb{E}_{\boldsymbol{t}\sim PF}f_{\boldsymbol{\theta_t}}(\boldsymbol{d}_i)\right)\\
&\leq \mathbb{E}\mathbb{P}(t_i=0)\mathcal{L}(f_{\boldsymbol{\theta}_t^F}(\boldsymbol{d}_i), c_i(\boldsymbol{d}_i)) + \mathbb{E}\mathbb{P}(t_i=1)\mathcal{L}\left(f_{\boldsymbol{\theta}_t^F}(\boldsymbol{d}_i), \mathbb{E}_{\boldsymbol{t}\sim PF}f_{\boldsymbol{\theta_t}}(\boldsymbol{d}_i)\right) + \mathbb{E}\mathcal{L}(f_{\boldsymbol{\theta}_t^{CF}}(\boldsymbol{d}_i), f_{\theta_t^F}(\boldsymbol{d}_i)).
\end{aligned}
\tag{6}
$$

Notice the fact

$$
\mathbb{E}_{\boldsymbol{t}\sim PF}\mathcal{L}(f_{\boldsymbol{\theta}_t^F}(\boldsymbol{d}_i), \bar{Y}_i) = \mathbb{E}\mathbb{P}(t_i=1)\mathcal{L}(f_{\boldsymbol{\theta}_t^F}(\boldsymbol{d}_i), c_i(\boldsymbol{d}_i)) + \mathbb{E}\mathbb{P}(t_i=0)\mathcal{L}\left(f_{\theta_t^F}(\boldsymbol{d}_i), \mathbb{E}_{\boldsymbol{t}\sim PF}f_{\boldsymbol{\theta_t}}(\boldsymbol{d}_i)\right).
\tag{7}
$$

Combined Eqn. 4 to Eqn. 7, our new objective $\min[\mathcal{L}_{PF}(\boldsymbol{\theta_t}) + \mathcal{L}_{PCF}(\boldsymbol{\theta_t})]$ can be transfered to:

$$
\underbrace{\mathbb{E}\mathcal{L}(f_{\boldsymbol{\theta}_t^F}(\boldsymbol{d}_i), c_i(\boldsymbol{d}_i))}_{\text{Fitting term}} + \underbrace{\mathbb{E}\mathcal{L}(f_{\boldsymbol{\theta}_t^{CF}}(\boldsymbol{d}_i), f_{\theta_t^F}(\boldsymbol{d}_i))}_{\text{Consistency term}} + \underbrace{\mathbb{E}_{\boldsymbol{t}\sim PF}\mathcal{L}(Y_i, \bar{Y}_i) + \mathbb{E}_{\boldsymbol{t}\sim PCF}\mathcal{L}(Y_i, \bar{Y}_i).}_{\text{Uniformity term}}
\tag{8}
$$

Via examining the upper bounds of the new loss function, we identify three terms that impact the effectiveness of few-shot NeRF, as shown in Eqn. 8 above. For the fitting item, it is approximately similar to the traditional loss as in Eqn. 4.2. It requires the network parameters under a certain sampling strategy can correct fitting to the ground truth. The consistency term requires that, under a certain sampling strategy, the optimal network parameters obtained from visible views and invisible views are as consistent as possible. This actually means that the distributions of visible views and invisible views should be consistent. It can be minimized by minimizing the distance between the representations of visible and invisible views. The uniformity term takes the minimum value under uniform sampling. It shows that the sampled views are to be distributed as evenly as possible. This upper bound guides us to capture new views to supplement the existing training set , which bring the most information. Moreover, the conclusion can not only guide sampling but also help the design of regularization terms because these properties are necessary conditions for an outstanding NeRF.

# 5 EXPERIMENTS

We verified our conclusions through experiments. First, we verified whether these conclusions can guide the sampling. Followed by ActiveNeRFPan et al. (2022), we provided several initial images and then supplemented the sampling of new views by guiding the sampling strategy to evaluate whether the sampling strategy can achieve the best performance with the least number of samples. The experiments showed that based on our sampling strategy, the best performance can be achieved with the same number of new views introduced. In addition, we designed regularization term loss functions based on these conclusions to constrain the training of NeRF, and the experiments showed that these regularization terms can significantly improve performance. The experiments prove that our framework is general and effective.

## 5.1 EXPERIMENTAL SETUP

**Datasets.** We extensively demonstrate our approach in two benchmarks, including Blender Mildenhall et al. (2021) and LLFF datasets Mildenhall et al. (2019). The Blender dataset contains 8 synthetic

objects with complicated geometry and realistic non-Lambertian materials. Each scene has 100 views for training and 200 for the test, and all the images are at 800×800 resolution. LLFF is a real-world dataset consisting of 8 complex scenes captured with a cellphone. Each scene contains 20-62 images with 1008×756 resolution, where 1/8 images are reserved for the test.

**Metrics.** We report the image quality metrics PSNR, SSIM and LPIPS for evaluations.

## 5.2 GUIDING SAMPLING UNDER A LIMITED INPUT BUDGET

**Setups.** We conduct experiments in Active Learning settings Pan et al. (2022), where the goal is to supplement the initial constrained training dataset with newly captured samples. We start by training a NeRF model using the initial observations. Then we render candidate views and estimate them using an acquisition function to select the most valuable ones. We proceed to further train the NeRF model using the newly acquired ground-truth images corresponding to these selected views.

**Design.** We adhere to the "train-render-evaluate-pick" scheme, and only modify the evaluation step. During this phase, we assess the semantic consistency between visible views and candidate views based on the consistency term. Additionally, we make the furthest view sampling based on the uniformity term. In our practical implementation, we use CLIP Radford et al. (2021) to extract semantic features from both ground truth images in the training set and rendered images in the candidate set. For a given candidate view, we measure the consistency term as the negative cosine similarity of semantic features and the uniformity term as the distance of camera position relative to the training set. So, our goal is to find a viewpoint that simultaneously has the minimum cosine similarity and the farthest camera position.

| Sampling Strategy | Setting I, 20 observations: | | | Setting II, 10 observations: | | |
|---|---|---|---|---|---|---|
| | PSNR ↑ | SSIM ↑ | LPIPS ↓ | PSNR ↑ | SSIM ↑ | LPIPS ↓ |
| NeRF + Rand | 16.626 | 0.822 | 0.186 | 15.111 | 0.779 | 0.256 |
| NeRF + FVS | 17.832 | 0.819 | 0.186 | 15.723 | 0.787 | 0.227 |
| ActiveNeRF | 18.732 | 0.826 | 0.181 | 16.353 | 0.792 | 0.226 |
| Ours (C→U) | 18.930 | **0.846** | **0.149** | 16.718 | **0.810** | **0.205** |
| Ours (U→C)) | **20.093** | 0.841 | 0.162 | **17.314** | 0.801 | 0.209 |

Table 1: Quantitative comparison in Active Learning settings on Blender. **NeRF + Rand:** Randomly capture new views in the candidates. **NeRF + FVS:** Capture new views using furthest view sampling. **ActiveNeRF:** Capture new views using ActiveNeRF scheme for Continuous Learning. **Ours (C→U):** First choose 20 views with the highest consistency term, and then capture views within them based on uniformity term. **Ours (U→C):** First capture 20 views with the highest uniformity term, and then capture views within them based on consistency term. **Setting I:** 4 initial observations and 4 extra observations obtained at 40K,80K,120K and 160K iterations. **Setting II:** 2 initial observations and 2 extra observations obtained at 40K,80K,120K and 160K iterations. 200K iterations for training in total. All results are reproduced by codebase provided by ActiveNeRF.

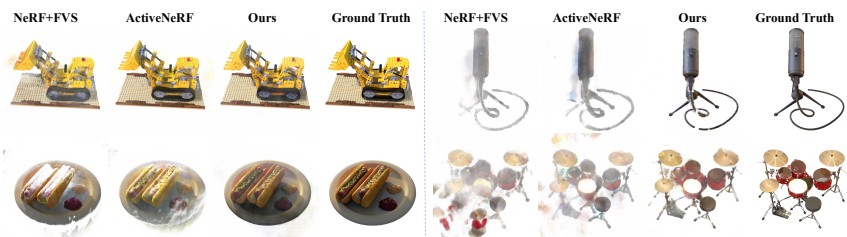

Figure 3: **Quantitative comparison in Active Learning settings on Blender.** Given limited input views, our strategy can select better candidate views. Our rendered images without excessively blurry boundaries exhibit greater clarity compared to those rendered by ActiveNeRF.

**Results.** We show the performance of our sampling strategy on Blender over baseline approaches in Table 1 and Figure 3. Our strategy outperforms baselines on the quality of view synthesis. It can be easily seen that our method, which takes into account the distribution consistency between

visible and invisible views along with a tendency towards uniform sampling, provides better guidance for sampling under a limited input budget. When we prioritize the consistency term before the uniformity term (*Ours (C→U)*), we observe higher LPIPS scores, which align more closely with human perception. Conversely, when first prioritizing the uniformity term (*Ours (U→C)*), we obtain higher PSNR scores, which are based on raw pixel value differences.

## 5.3 REGULARIZATION TERM FOR GENERAL NERF

**Setups.** To demonstrate the effectiveness of our method, we evaluate our method on two datasets under few-shot settings: the NeRF Blender Synthetic dataset (Blender) Mildenhall et al. (2021), the LLFF datasetMildenhall et al. (2019). We follow the few-shot settings with previous work and make a fair comparison with them. For Blender, we follow DietNeRF Jain et al. (2021) to train on 8 views and test on 25 test images. For LLFF, we use the exact same protocol as RegNeRF and FreeNeRF. We report PSNR, SSIM, and LPIPS scores as quantitative results.

**Design.** Due to the consistency term requiring consistency between visible and invisible views, we demand semantic consistency between the images from the training set and the images obtained by NeRF parameters. Meanwhile, the uniformity term requires sampling to be diverse, meaning that the images from the training set and NeRF parameters should have enough variance. Thus, we require a sufficiently large variance in the pixel-wise mean value between visible and invisible images. In conclusion, the consistency term expects a NeRF to ensure semantic consistency between visible and invisible views, while the uniformity term aims to maintain overall semantic consistency while maximizing diversity.

**Dataset.** The Blender dataset Mildenhall et al. (2021) has 8 synthetic scenes in total. We follow the data split used in DietNeRF Jain et al. (2021) to simulate a few-shot neural rendering scenario. For each scene, the training images with IDs (counting from "0") 26, 86, 2, 55, 75, 93, 16, 73, and 8 are used as the input views, and 25 images are sampled evenly from the testing images for evaluation. LLFFMildenhall et al. (2019) is a dataset containing a total of 8 scenes. Consistent with RegNeRF and FreeNeRF, we used every 8th image as a new view for evaluation and sampled 3 input views evenly across the rest of the views.

| Method | PSNR ↑ | SSIM ↑ | LPIPS ↓ |
|---|---|---|---|
| NeRF | 14.934 | 0.687 | 0.318 |
| NV | 17.859 | 0.741 | 0.245 |
| Simplified NeRF | 20.092 | 0.822 | 0.179 |
| DietNeRF | 22.503 | 0.823 | 0.124 |
| **DietNeRF+Ours** | 22.860(+0.360) | 0.859(+0.036) | 0.114(-0.01) |
| FreeNeRF | 24.259 | 0.883 | 0.098 |
| **FreeNeRF+Ours** | **24.896(+0.637)** | **0.894(+0.011)** | **0.096(-0.002)** |

Table 2: Quantitative comparison on Blender.

**Results on Blender dataset.** Table 2 shows the result on the Blender dataset. For all methods we can directly introduce consistency term and uniformity term, to verify the usefulness of our proposed framework. For DietNeRF, the consistency loss actually belongs to the consistency term, so DietNeRF is a degradation of our framework. We add uniformity term to its original design, and the results show that with uniformity term, we can form good constraints on the distribution of unobserved samples to make it as diverse as possible. For FreeNeRF, since frequency loss does not belong to consistency term and uniformity term, we additionally add these two items as constraints. The results show that our framework can also help FreeNeRF to improve the results significantly. With the help of our framework, we can better design and use various types of regularization terms to improve the results. At the same time, it also inspires us to what perspective we should consider to design new regularization terms.

**Results on LLFF dataset.** Table 3 and Figure 4 shows the quantitative and qualitative result on the LLFF dataset. With the help of uniformity term and consistency term, we can further improve FreeNeRF by a big margin. It is noted that all numbers are directly borrowed from FreeNeRF. Transfer learning-based methods perform much worse than ours on the LLFF dataset due to the non-trivial domain gap between DTU and LLFF. Compared to FreeNeRF, our method constrains the distribution

| Method | Setting | PSNR ↑ | SSIM ↑ | LPIPS ↓ |
|---|---|---|---|---|
| SRF | | 12.34 | 0.250 | 0.591 |
| PixelNeRF | Trained on DTU | 7.93 | 0.272 | 0.682 |
| MVSNeRF | | 17.25 | 0.557 | 0.356 |
| SRF ft | Trained on DTU | 17.07 | 0.436 | 0.529 |
| PixelNeRF ft | and | 16.17 | 0.438 | 0.512 |
| MVSNeRF ft | Optimized per Scene | 17.88 | 0.584 | 0.327 |
| Mip-NeRF | | 14.62 | 0.351 | 0.495 |
| DietNeRF | Optimized per Scene | 14.94 | 0.370 | 0.496 |
| RegNeRF | | 19.08 | 0.668 | 0.283 |
| mip-NeRF concat. | | 16.11 | 0.401 | 0.460 |
| RegNeRF concat. | Optimized per Scene | 18.84 | 0.573 | 0.345 |
| FreeNeRF | | 19.63 | 0.612 | **0.308** |
| **FreeNeRF + Ours** | Optimized per Scene | **20.02(+0.39)** | **0.616(+0.004)** | 0.318(+0.01) |

Table 3: Quantitative comparison on LLFF.

of observable and unobservable views, and the variance of the unobservable distribution. This can lead to new constraints on both consistency and diversity for unobservables, and the improvement of the results demonstrates the effectiveness of our method. For instance, in the "flower" example in Figure 4, our methods can better distinguish the outline of the flowers from the background. Consistency term allows NeRF to better maintain the consistency of the geometric outline, and uniformity term enables NeRF to render various flowers from different perspectives.

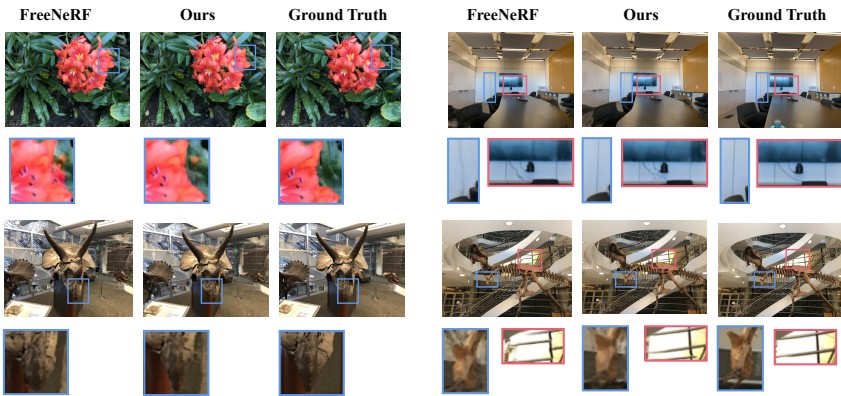

Figure 4: **Qualitative comparison on LLFF.** Given 3 input views, we show novel views rendered by FreeNeRF and ours on the LLFF dataset. Compared with FreeNeRF, our method can provide better geometry for the observed objects. For the "horns" and "trex" examples, FreeNeRF fails to render sharp outlines in some places, but our additional losses can gain more detailed skeleton structure.

## 6 CONCLUSION

In this paper, we revisit the novel view synthesis via a casual perspective. Inspired by causal representation learning, we reformulate the task loss by ITE(Identification Treatment Effect). Then we propose a framework that theoretically support the main idea. We derived the upper bound of the ITE loss, which can be divided into three terms. The fitting term requires the optimal network parameters under a certain view sampling strategy to support correct fitting to the ground truth. The consistency term requires that the distributions of visible views and invisible views should be consistent. The uniformity term require the sampled views to be distributed as evenly as possible. This upper bound can guide us how to capture new views to supplement the existing training set, which bring the most information gain. The conclusion can not only guide sampling but also help the design of regularization terms. Finally, we have validated our conclusion experimentally.

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
