# Exploring View Sampling Strategy in Novel View Synthesis from Causal Perspectives

## 1 GUIDING SAMPLING UNDER A LIMITED INPUT BUDGET

We conduct experiments in Active Learning settings using the ActiveNeRF Pan et al. (2022) codebase. In traditional NeRF Mildenhall et al. (2021), we obtain a volume parameter $\sigma$ and color values $c = (r, g, b)$ for a specific position and direction. In ActiveNeRF, it simultaneously outputs both mean and variance, following a Gaussian distribution. For simplicity, we adopt the ActiveNeRF version and apply its pipeline to our baseline methods *(NeRF+Random, NeRF+FVS)* as well as our proposed strategy. The primary modification we make is in the evaluation step, which is central to this active learning setting.

Its original codebase only provides training configuration files for a portion of the LLFF dataset and the Blender dataset. We observe that for the Blender dataset, the codebase used a fixed number (20) of initial training samples so we cannot decide the initial training set size. We then modify it to allow the selection of the initial training set size, with the remaining images serving as a holdout set. For instance, in Setting I, for each object in the Blender dataset with 100 ordered images, we choose the first 4 images as the initial set and use the remaining 96 images as the holdout set. Due to excessive memory requirements, training on the LLFF dataset is not feasible even on a 48GB A40 GPU, so we temporarily refrain from conducting experiments on it. However, we believe that the results on the Blender dataset sufficiently validate our claims.

Due to the randomness of the strategy and potential variations in the training process, we conducted three experiments for each result and selected the best outcome. In Table 1, We provide a detailed breakdown of the specific results for each object on Blender in Setting I.

| PSNR↑ | hotdog | lego | chair | drums | ficus | materials | mic | ship | Avg. |
|---|---|---|---|---|---|---|---|---|---|
| NeRF + Rand | 22.19 | 19.85 | 19.99 | 10.93 | 18.13 | 8.73 | 17.85 | 15.31 | 16.62 |
| NeRF + FVS | 23.87 | 17.83 | 20.06 | 15.38 | 17.91 | 13.76 | 17.91 | 15.94 | 17.83 |
| ActiveNeRF | 17.87 | 18.96 | 20.20 | 14.82 | **22.55** | **18.19** | 17.92 | **19.34** | 18.73 |
| Ours (C→U)) | **24.01** | 20.48 | **26.21** | 16.78 | 18.49 | 13.95 | 17.57 | 13.95 | 18.93 |
| Ours (U→C) | 23.14 | **22.90** | 20.08 | **17.96** | 20.99 | 15.16 | **24.01** | 16.50 | **20.09** |
| **SSIM↑** | hotdog | lego | chair | drums | ficus | materials | mic | ship | Avg. |
| NeRF + Rand | 0.919 | 0.838 | 0.848 | 0.793 | 0.845 | 0.762 | 0.881 | 0.689 | 0.822 |
| NeRF + FVS | **0.922** | 0.798 | 0.853 | 0.776 | 0.838 | 0.776 | 0.879 | 0.706 | 0.819 |
| ActiveNeRF | 0.860 | 0.829 | 0.858 | 0.768 | **0.886** | **0.813** | 0.876 | 0.716 | 0.826 |
| Ours (C→U)) | 0.918 | **0.852** | **0.898** | 0.793 | 0.848 | 0.789 | 0.883 | **0.789** | **0.846** |
| Ours (U→C) | 0.916 | 0.851 | 0.849 | **0.814** | 0.859 | 0.812 | **0.924** | 0.704 | 0.841 |
| **LPIPS↓** | hotdog | lego | chair | drums | ficus | materials | mic | ship | Avg. |
| NeRF + Rand | 0.089 | 0.152 | 0.165 | 0.231 | 0.152 | 0.241 | 0.138 | 0.317 | 0.186 |
| NeRF + FVS | **0.082** | 0.197 | 0.158 | 0.239 | 0.167 | 0.205 | 0.140 | 0.304 | 0.186 |
| ActiveNeRF | 0.172 | 0.150 | 0.149 | 0.253 | **0.116** | **0.145** | 0.142 | 0.319 | 0.181 |
| Ours (C→U)) | 0.089 | **0.135** | **0.109** | 0.218 | 0.152 | 0.177 | 0.139 | **0.177** | **0.149** |
| Ours (U→C)) | 0.099 | 0.153 | 0.165 | **0.183** | 0.136 | 0.159 | **0.093** | 0.306 | 0.162 |

Table 1: **Quantitative comparison on Blender in Setting I.** We provide a detailed listing of the metric values for each object on Blender, which is the same in Table 1 in the manuscript.

## 2 REGULARIZATION TERM FOR GENERAL NERF ON DTU.

Figure 1: **Example of our results with 3 input views on the DTU dataset.**

| Method | Object | | Full-image | |
|---|---|---|---|---|
| | PSNR ↑ | SSIM ↑ | PSNR ↑ | SSIM ↑ |
| SRF Chibane et al. (2021) | 15.32 | 0.671 | 15.84 | 0.532 |
| PixelNeRF Yu et al. (2021) | 16.82 | 0.695 | **18.74** | 0.618 |
| MVSNeRF Chen et al. (2021) | 18.63 | 0.769 | 16.33 | 0.602 |
| SRF ft Chibane et al. (2021) | 15.68 | 0.698 | 16.06 | 0.550 |
| PixelNeRF ft Yu et al. (2021) | 18.95 | 0.710 | 17.38 | 0.548 |
| MVSNeRF ft Chen et al. (2021) | 18.54 | 0.769 | 16.26 | 0.601 |
| Mip-NeRF Barron et al. (2021) | 8.68 | 0.571 | 7.64 | 0.227 |
| DietNeRF Jain et al. (2021) | 11.85 | 0.633 | 10.01 | 0.354 |
| RegNeRF Niemeyer et al. (2022) | 18.89 | 0.745 | 15.33 | 0.621 |
| mip-NeRF concat. (repro.) | 9.10 | 0.578 | 7.94 | 0.235 |
| RegNeRF concat. (repro.) | 18.50 | 0.744 | 15.00 | 0.606 |
| FreeNeRFYang et al. (2023) | 19.92 | **0.787** | 18.02 | 0.680 |
| **FreeNeRF + Ours** | **20.11(+0.19)** | 0.785(-0.002) | 18.41(+0.39) | **0.681(+0.001)** |

Table 2: **Quantitative comparison on DTU.** We follow the experiment setting in FreeNeRF and present the PSNR and SSIM scores of foreground objects and full images. Compared with FreeNeRF and other baselines, We can observe that ours based on FreeNeRF can better synthesize foreground objects and full images, especially in PSNR.

In 3-view setting, we also conduct additional experiments on the DTU datasetJensen et al. (2014) following the setting of FreeNeRF. It contains 124 scenes and we follow Niemeyer et al. (2022) to optimize NeRF models directly on the 15 test scenes. The test scan IDs are: 8, 21, 30, 31, 34, 38, 40, 41, 45, 55, 63, 82, 103, 110, and 114. In each scan, the images with the following IDs (counting from "0") are used as the input views: 25, 22, 28, 40, 44, 48, 0, 8, 13. We use the first 3 images as the input views in a 3-view setting. The 25 images with IDs in [1, 2, 9, 10,11, 12, 14, 15, 23, 24, 26, 27, 29, 30, 31, 32, 33, 34, 35, 41, 42, 43, 45, 46, 47] serve as the novel views for evaluation. We follow Niemeyer et al. (2022); Yu et al. (2021) to use a 4× downsampled resolution, resulting in 300 × 400 pixels for each image.

Table 2 and Figure 1 show quantitative and qualitative results on the DTU dataset. We find that masks of the DTU dataset do not always help improve PSNR ans SSIM and sometimes the PSNR score in a

specific scene drops a lot. For a fair comparison, we train one model for one scene to produce the results in the object and full-image setting at the same time. Transfer learning-based methods Chibane et al. (2021); Yu et al. (2021); Chen et al. (2021) that require expensive pre-training underperform ours in almost all settings, except the full-image PSNR score of Yu et al. (2021). This may be due to the bias introduced by the white table and black background present in many scenes in the DTU dataset. Compared with FreeNeRF, our method can get better performance in the full-image setting.