# OpenReview forum: "Exploring View Sampling Strategy in Novel View Synthesis from Causal Perspectives"
_ICLR.cc/2024/Conference — Submitted to ICLR 2024_

### Official Review · Reviewer_HZXU · 2023-10-29

**Soundness:** 2 fair
**Presentation:** 2 fair
**Contribution:** 2 fair
**Rating:** 5
**Confidence:** 3

**Summary:**

This paper studies how to train NeRF with the optimal training set under limited view inputs for novel view synthesis. It proposes a theoretical framework for view sampling strategies from a causal perspective, finally decomposing the objective into three components: a fitting term similar to traditional NeRF training loss, a consistency term requiring consistency between visible and invisible views, and a uniformity term demanding the sampling to be diverse. The proposed sampling strategy induces higher-quality NeRFs and can be used as regularization term for general NeRF training.

**Strengths:**

1. Framing the novel view synthesis problem via a causal perspective is novel.
2. The deduced supervision objective with three terms is intuitive and well-explained.
3. Experiments demonstrate that based on the proposed sampling strategy better performance could be achieved with the same number of training views, using the principles as a regularization term to the training of general term could also improve performance.

**Weaknesses:**

1. Although the derived supervision objective is intuitive, the framing of novel view synthesis problem with causal framework is a bit obscure with mistakes: e.g. page 5 the authors mentioned "we defer the details to the Appendix" which do not exist, Eq. 4 in page 6 is also falsely rendered.
2. Two variants of the model are proposed (prioritizing consistency and uniformity term differently) without a consistency in which one would perform better which may limit the usability.

**Questions:**

1. in Appendix Tab. 1, ActiveNeRF acquires better results 3/8 on ficus, materials and ship, are there any explainations for this?
2. Could some qualitative comparisions with DietNerF (which would show the effects of uniformity loss only) be povided ?

---

### Official Review · Reviewer_itVg · 2023-10-29

**Soundness:** 3 good
**Presentation:** 3 good
**Contribution:** 1 poor
**Rating:** 5
**Confidence:** 2

**Summary:**

The authors introduced a view sampling strategy for novel view synthesis, grounded in the perspective of causal representation learning. They identified three key metrics to assess sampling performance: the fitting term, the consistency term, and the uniformity term. Additionally, they presented a novel theoretical framework addressing the sampling challenge within NeRF.

**Strengths:**

1. The introduction of the causal perspective in the view sampling algorithm holds significant potential and could serve as a foundational approach for future research in this domain.
2. The authors meticulously lay out a comprehensive mathematical framework that not only elucidates the underlying problem but also leads to the derivation of the three pivotal terms central to their methodology.
3. The paper stands out for its clarity and coherence, ensuring that readers, regardless of their expertise level, can grasp the concepts and findings presented."

**Weaknesses:**

1. The rationale behind the view-sampling task raises questions. In certain scenarios, acquiring additional view images can be challenging. However, when a substantial number of dense views are already available, the motivation to devise a sampling strategy for training the neural rendering model with sparse views appears insufficient. Specifically, the activeNeRF model's primary objective is to identify the most optimal camera view for capturing the training image, rather than selecting from a plethora of pre-existing images.
2. The paper's primary contribution seems to be the introduction of a metric or loss function to evaluate the selected views. However, the absence of an ablation study that separately assesses the impact of each of these three terms is a missed opportunity for deeper understanding. As a result, the contribution feels somewhat lacking in depth.
3. The proposed loss function presents challenges in differentiability with respect to 't'. The sampling proposal, derived from the farthest sampling strategy, may not be the most efficient approach. It appears to demand significant training resources, resulting in elevated training costs. The potential enhancements in model performance might not justify the trade-off in terms of the increased training time and resource allocation.

**Questions:**

Please see the weakness above.

---

### Official Review · Reviewer_bNPg · 2023-10-30

**Soundness:** 2 fair
**Presentation:** 1 poor
**Contribution:** 2 fair
**Rating:** 3
**Confidence:** 3

**Summary:**

This paper studies the view sampling strategies of Nerf reconstruction from a causal perspective. The authors try to solve the problem using a small subset of photos from a total of K potential views, to achieve the best reconstruction. To solve this, the authors propose to use causal represntation learning using loss by Identification Treatment Effect. They propose three terms, a normal fitting term as reconstruction loss, a consistency term to ensure consistency between visible views and invisible views and a uniformity term requires the samples to be distributed evenly. The results show the proposed strategy can provide slightly better reconstruction compared to alternative baselines in the proposed setting.

**Strengths:**

* The paper proposes a novel perspective to study the view sampling problem in volumetric reconstruction using NeRF as an example. This take-away can potentially also generalize other multiview reconstruction algorithms.
* Given its current setting, the hypothesis is validated on nerf reconstruction datasets, with small improvement compared to its baselines.

**Weaknesses:**

* The presentation of this paper could be greatly improved. I may not have understand a lot of details correctly given its current presentation.
  * It is very hard to read without being very familiar with ActiveNeRF and casual representation learning. Have to trace to original papers for more details. This could be added to the preliminary parts.
  * Too many notations which makes things more complicated than needed. I don't think I found how exactly the loss of consistency term and uniformity term were calculated in (8) at runtime. As I understand, the method should be as simple as calculating the reconstruction loss using different groups of input samples. Provide an algorithm chart of how of how P^{F}, P^{hat}^{CF} and P^{CF} will greatly help.
  * There are some notations introduced in 4.1 (e.g. P(Y|do(d))) are not explained until 4.2.
* Overall I am not sure I understand the real-world impact of this paper using the proposed strategy. Maybe I had some misunderstanding in the details given my concern on its presentation. Please correct me if I am wrong here. The goal of this paper to find "optimal sampling strategy for training set", "K_s corresponding photos as sparse sample inputs among K_d total potential views" is hardly a real problem statement for its real-world use case, which is my biggest concern for this proposed application of causal representation learning. From sampling perspective, we can use all the K_d potential views as long as they are available. As I understand, the evaluation of the counter factual distribution will require using the non-selected but captured images as supervision, which is not how active learning is executed in real-world case. Given this setting, it makes the results also less appealing in contrast to alternative baselines (which learns to predict next-best unknown view) given the fact all images from that particular datasets are used in evaluating the sampling strategy.

**Questions:**

1. My major question is around how the clarity of the sampling process in training time. Confirm any places I misunderstood about this paper, as I highlighted in the weakness part.
2. I am also curious how the views are sampled finally for different groups in the final results. Provide some visualization and discussions about them can be very helpful to guide the view-sampling process in real world applications. I wonder how that indicate the connection of uniformity term and consistency term are correlated to the camera FoV and ray distributions.

---

### Meta-Review · Area_Chair_sHkQ · 2023-12-05

**Metareview:**

This paper proposes a causal learning approach to efficient sampling of views for NeRF-style generation. Reviewers generally agreed that the area is important and that the authors are proposing something interesting. However, all reviewers found the paper, including its core ideas, not clear enough to effectively evaluate. I agree. The authors did not provide a rebuttal, so many of these questions have not been answered.

I believe once the authors rewrite and clarify the paper, taking the suggestions and questions raised by the reviewers into account, it will have a solid shot at clearing the bar.

**Justification For Why Not Higher Score:**

General lack of clarity in the paper prevents it from reaching the bar.

**Justification For Why Not Lower Score:**

N/A

---

### Decision · Program_Chairs · 2024-01-16

Reject